# Experimental and Modeling Study on Cr(VI) Migration from Slag into Soil and Groundwater

**Xiange Wu** [1], **Tiantian Ye** [2], **Chunsheng Xie** [1], **Kun Li** [3,4], **Chang Liu** [2], **Zhihui Yang** [5], **Rui Han** [6], **Honghua Wu** [7] and **Zhenxing Wang** [2,*]

1    Guangdong Provincial Key Laboratory of Environmental Health and Land Resource, School of Environmental and Chemical Engineering, Zhaoqing University, Zhaoqing 526061, China
2    South China Institute of Environmental Sciences, Ministry of Ecology and Environment, Guangzhou 510655, China
3    Freeman Business School, Tulane University, New Orleans, LA 70118, USA
4    Guangzhou Huacai Environmental Protection Technology Co., Ltd., Guangzhou 511480, China
5    School of Metallurgy and Environment, Central South University, Changsha 410083, China
6    China Environment Publishing Group, Beijing 100062, China
7    Guangxi Zhengze Environmental Protection Technology Co., Ltd., Hezhou 542800, China
\*    Correspondence: wangzhenxing@scies.org

**Abstract:** The transport and prediction of hexavalent chromium (Cr(VI)) contamination in "slag–soil–groundwater" is one with many uncertainties. Based on the column experiments, a migration model for Cr(VI) in the slag–soil–groundwater system was investigated. The hydraulic conductivity (Kt), distribution coefficient (Kd), retardation factor (Rd), and other hydraulic parameters were estimated in a laboratory. Combining these hydraulic parameters with available geological and hydrogeological data for the study area, the groundwater flow and Cr(VI) migration model were developed for assessing groundwater contamination. Subsequently, a Cr(VI) migration model was developed to simulate the transport of Cr(VI) in the slag–soil–groundwater system and predict the effect of three different control programs for groundwater contamination. The results showed that the differences in the measured and predicted groundwater head values were all less than 3 m. The maximum and minimum differences in Cr(VI) between the measured and simulated values were 1.158 and 0.001 mg/L, respectively. Moreover, the harmless treatment of Cr(VI) slag considerably improved the quality of groundwater in the surrounding areas. The results of this study provided a reliable mathematical model for transport process analysis and prediction of Cr(VI) contamination in a slag–soil–groundwater system.

**Keywords:** Cr(VI); groundwater; migration; model; slag; soil

## 1. Introduction

Chromium (Cr) contamination of soil and groundwater is a pervasive concern of the scientific and social community in recent years because of its high toxicity [1], solubility, and mobility [2,3]. Cr, especially hexavalent Cr (Cr(VI)), is considered toxic to humans and the environment with sufficient evidence [4,5]. With the rapid development of industrialization, soil and groundwater worldwide have been contaminated by solid and liquid Cr(VI) wastes [6,7]. Consequently, the restoration of soil and groundwater has become a significant concern in most waste sites [8,9]. Moreover, the assessments of the migration and fate of Cr(VI) in soil and groundwater are vital to achieving relevant environmental restoration goals [10,11].

Several studies have thoroughly assessed Cr(VI) migration through soil or groundwater in mining soil [12,13], in a tannery landfill [14,15], and in a chromate production site [16,17]. For example, Jardine et al. investigated the impact of coupled hydrologic and geochemical processes on the transport of Cr(VI) in undisturbed heterogeneous soil [18].

Using a multireaction transport model to evaluate Cr(VI) transport, they found that the transport of Cr(VI) mass in undisturbed heterogeneous soil featured the nonlinear nature of the adsorption process coupled with irreversible sorption kinetics. Meanwhile, Rao et al. utilized available hydrogeological, geophysical, and groundwater quality data to build a groundwater flow and Cr(VI) transport model [19]. Using this model, they predicted the extent of the total dissolved solids in groundwater for loading conditions at the dump site over 30 years. In another study, the migration and species distribution of Cr and inorganic ions from tanneries in soils and groundwater were analyzed, and the influence of Cr precipitation or dissolution was related to the source strength, coexisting ions, and pH [20].

To date, limited effort has been directed toward the assessment of Cr(VI) migration through independent systems (slag, soil, or groundwater) and less toward comprehensive consideration of the complex pollution sources, rainfall, soil, groundwater, and other pollution pathways [21,22]. However, in actual cases, the Cr-containing slag in opening yards inevitably results in the overall pollution of soil and groundwater. Therefore, in this research, a typical factory in Xiangxiang City, Hunan Province, China, was taken as the study object, and the pollution of Cr(VI) from Cr-containing slag to the soil and groundwater ("slag–soil–groundwater" system) was investigated. In our previous study, using artificial neural network and genetic algorithm methods, a model was built and applied to estimate the total Cr(VI) released from the slag into the soil under the effects of rainfall [23]. Then, with the adsorption and transport parameters of Cr(VI) in soil determined by batch and column experiments, the model of Cr(VI) transport through the "slag–soil" system was established. The HYDRUS-1D dynamics model was found to effectively simulate the Cr(VI) migration in soil [24]. In this work, on the basis of geology and hydrogeology of the study area, the migration of Cr(VI) in groundwater was simulated using the Cr(VI) migration model with the results of the Cr(VI) leaching transport in soil. Lastly, a mathematical model of the overall migration kinetics quantitatively describing the Cr pollution in the "slag–soil–groundwater" system was constructed to achieve quantitative estimation and effective prediction of historical and future pollution.

## 2. Materials and Methods

### 2.1. Geology and Hydrogeology of the Study Area

Xiangxiang City, China, has a subtropical moist monsoon climate, with plenty of rainfall and high plateaus. Since 1965, the iron alloy factory in Xiangxiang City has used Cr ore to produce steel alloy, silicomanganese alloy, and Cr–silicon alloy, among others. Simultaneously, the Cr-containing slag has been randomly stacked in the open yard near the factory. The leachate generated from the slag infiltration by rainfall has been polluting the soil and groundwater.

The study site is characterized by a gentle terrain, with a 2° to 5° slope, classified as a type of flood terrace terrain with a 1 to 5 km width. The geological structure is composed of loose Quaternary deposits and tertiary red rock (Figure 1). The components of the loose Quaternary deposits include a sand layer, sub-sand layer, a sub-sandy soil, and gravel stones. The total thickness of the layers is 4 to 27 m. The compressive strength of Quaternary red clay and sandy clay is generally 20 t/m$^3$. Adverse geological phenomena, such as caving, collapse, landslide, and subsidence, were not observed. The sandy gravel layer contains abundant pore water, with very shallow groundwater, only a few centimeters to tens of centimeters above the surface. The pH is about 6.7–7.9. The salinity of the groundwater is about 0.25 g/L. The water quality is of HCO$_3$–Ca and HCO$_3$–(Ca + Mg) types, making it a good source of drinking water. The tertiary red rock consists of mud–sandy cement with a thickness of hundreds of meters and is a good aquifer without fracture development, serving as a barrier that prevents the pollution of deep groundwater. More information about the study site was presented in our previous publication [23,24].

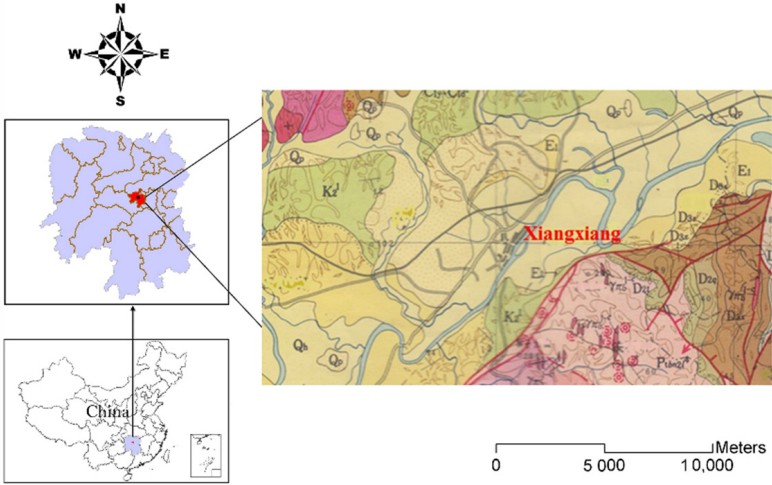

**Figure 1.** Geology of the study area.

## 2.2. Sample Preparation and Device

The sampling and characterizations of the soil selected for this study are presented in our previous work [24]. Soil was naturally air-dried in the laboratory and then passed through a 4-mesh sieve of 6 mm hole size to remove large quantities of sand, gravel, and debris. An 8-mesh sieve of 2 mm hole size was used to further remove minute soil samples. The soil samples with particle sizes between 2 and 6 mm were compacted layer by layer and then placed into a soil column. The bulk density was controlled to 1.64 g/cm$^3$ to approximate the bulk density of the underground water zone in undisturbed soil.

According to international and domestic research, an experimental device that ensures the dynamic simulation of pollutant migration with a constant water level was designed (Figure 2). The experimental device includes an organic glass column, submerged pumps, a liquid storing barrel, and a water tank with a constant water level, a water level relay, a bracket, sampling tubes, catheters, and a valve. Positioned horizontally, the soil column has a 700 mm effective length and 95 cm$^2$ internal cross-sectional areas. Three sampling holes were established at 200, 400, and 600 mm away from the entrance of the soil column. These holes were designated as points 1, 2, and 3, respectively. Water outlets were set at the terminal, with each water outlet blocked with a rubber stopper. The water outlets were connected with latex tubes to facilitate sampling. During the sampling, the supernatant of leaching procedure was filtered with a filter paper. The Cr(VI) concentration of the filtrate was determined using the diphenylcarbazide colorimetric method according to the standard method of China (GB 7467-87) and the detection limit was 4 μg/L.

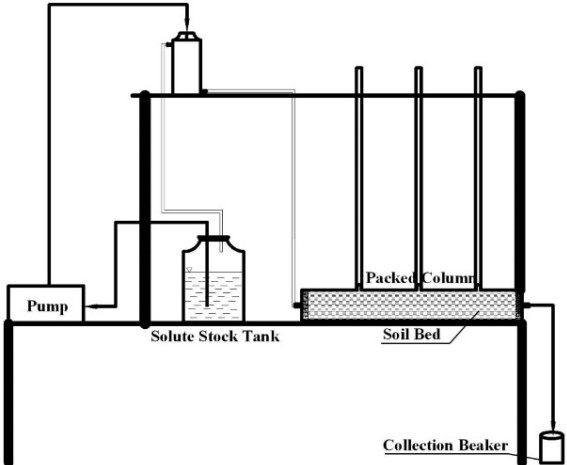

**Figure 2.** Schematic of the experimental equipment for contaminant transport in groundwater.

*2.3. Column Experiments*

2.3.1. Permeation Experiment

The distilled water in the liquid storage barrel was sent to the tank with a constant water level through a peristaltic pump. The constant water level was controlled to a vertical height of 810 mm from the center of the soil column. The soil column was leached with distilled water for a long period to induce ion leaching in the soil and achieve balance in the soil–water phase. When the water levels in sampling tubes 1, 2, and 3 remained invariant, the soil column reached saturation. When the water flow in the terminal outlet was stable, data were recorded.

2.3.2. Experiment on the Migration Characteristics of Cr(VI) in Groundwater

A total of 0.05 mol/L of Cr(VI) (200 mg/L) in NaCl solution was added to the original preparation solution; that is, 1775 mg/L of chloride ions (Cl$^-$) was used in the experiment. The experiment was conducted under continuous water flow saturation conditions at a constant water level of 810 mm. The results of the permeation experiment showed a small hydraulic conductivity of porous media. Obtaining water samples points from 1, 2, and 3 while satisfying the analysis requirements was difficult. Thus, sampling was restricted to one sampling tube. Samples were taken every 30 min (sampling water level: 570 mm), and the Cr(VI) and Cl$^-$ concentrations were determined. Water sampling was discontinued when the concentration of various solutes reached a balance.

*2.4. Groundwater Flow and Cr(VI) Migration Modeling*

2.4.1. Conceptualization of the Model

Determining the Scope for the Simulated Region

The range of the simulative calculations included the Lianshui River in the southeast, the water line (70 m) in the groundwater near the Xiangqian railway on the north side, and 2.8 km from the enterprise in the northeast and west. The total area for the simulation was about 20 km$^2$. The terrain data on the study site were provided by the Hunan Province Geology Survey Institute.

Unit Subdivision

In the contour plan, the minimum subdivision scales were 30 × 30 m for the slag field and 60 × 60 m for the other regions. In the vertical direction, the study site was divided into two layers. The first was the unconfined aquifer layer, and the second was the aquifuge. The range of the simulation height was about 20 m. The shallow groundwater was buried mainly in the sandy gravel layer at the bottom of the Quaternary red soil layer. The geological drilling data showed that the thickness of the sandy gravel layer was mainly 4 to 10 m, whereas the thickness of the local sections reached 14.20 m. In this study, therefore, the unconfined aquifer layer was set to 10 m.

Generalization of Aquifer Type

According to its type, lithology, thickness, and hydraulic conductivity, the aquifer should be heterogeneous. However, the simulation model was generalized as a homogeneous isotropic aquifer because its scope was limited and the simulation object was groundwater in an unconfined aquifer layer.

Generalization of Groundwater Flow Type

The terrain in the study area is a gentle slope, and the unconfined aquifer layer is distributed over the entire area, belonging to a laminar flow movement. The water flow migration that accords with Darcy's law disregards the effect of dry and wet periods on the groundwater level. The entire region thus be regarded as characterized by a stable dimensional plane flow.

Classification of Boundaries in the Study Site

Vertical boundary: The upper boundary is a water table, serving as a boundary for water exchange. The lower boundary is composed of tertiary red rock and a regional aquifuge, which is generalized as an aquifuge boundary.

Lateral boundary: The stream line serves as a boundary on the northeast, west, and southwest sides. It was set as a zero-flux impervious boundary. The Lianshui River in the southeast is a river boundary with a known water table. No natural boundary was found in the northern part, but a designated water head boundary can be identified on the basis of long-term observations of the water line (Figure 3).

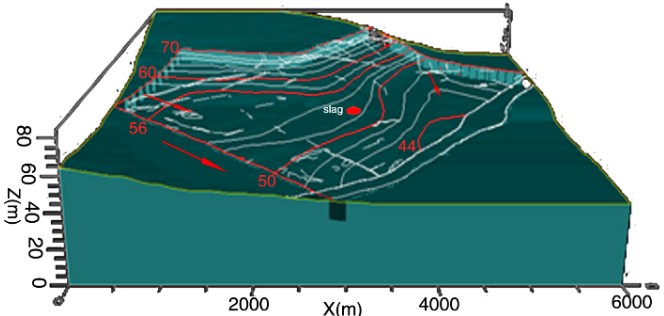

**Figure 3.** Terrain, boundary, and surveyed groundwater head distribution of the study area.

Solute boundary: For the solute boundary, the Cr(VI) slag field was set as a solute flux boundary in our simulation. The soil migration simulation results in previous studies of Cr(VI) transport through the soil were combined with those of the solute flux boundary simulation to determine the Cr(VI) concentration and realize solute flux. Specifically, the concentration of leached Cr(VI) from the bottom of the soil was calculated through soil migration simulation for every time interval. The results were then introduced into the pollutant flux setting in the groundwater simulation.

Computation of Groundwater Source Pooling

The recharge of the groundwater in the study site results primarily from precipitation and the seasonal lateral recirculation supply of the Lianshui River to the groundwater during flooding. This paper focused on precipitation recharge. Groundwater is consumed primarily through evaporation and water pumping from wells; evaporation loss was another focus of this study. The amounts of precipitation penetration and evaporation were controlled using the precipitation penetration coefficient and diving evaporation coefficient. The precipitation data were provided the Xiangxiang Meteorological Agency. The amount of precipitation penetration recharge can be calculated as follows:

$$R_{\text{prec}} = P \times a_{\text{prec}} \tag{1}$$

where $R_{\text{prec}}$ is the precipitation penetration recharge amount (mm), $P$ is the annual precipitation amount (mm), and $a_{\text{prec}}$ denotes the ratio of the precipitation penetration amount to the total precipitation amount, called the precipitation penetration coefficient. The precipitation penetration coefficient is affected mainly by the lithology and structure of the topsoil layer, precipitation amount, precipitation type, terrain slope, vegetation cover, and other factors, and the influence of the lithology of the topsoil layer is usually the most significant [25]. In the study site, the water level is shallow, the penetration is strong, and precipitation immediately penetrates the aquifer. Precipitation penetration is the main source of water supply, according to the dynamic long-term groundwater observation data and the hydrological calculation standard of water conservancy and hydropower projects issued by the Ministry of Water Resources of PRC. The $a_{\text{prec}}$ value was set to 0.17 for the region.

In addition, combined with the hydrogeological conditions of the enterprise in the region, according to the hydrological calculation standard of water conservancy and hydropower projects, the diving evaporation coefficient and diving limit evaporation depths were set to 0.4 and 4 m, respectively.

2.4.2. Groundwater Flow Equation

Before the establishment of the Cr(VI) migration model, the groundwater flow and contaminant migration equation must be determined. The groundwater flow equation for three-dimensional contaminant transport in porous media can be written as [26]:

$$S_r \frac{\partial H}{\partial t} = \frac{\partial}{\partial x}\left(K_{xx}\frac{\partial H}{\partial x}\right) + \frac{\partial}{\partial y}\left(K_{yy}\frac{\partial H}{\partial y}\right) + \frac{\partial}{\partial z}\left(K_{zz}\frac{\partial H}{\partial z}\right) \tag{2}$$

where $K_{xx}$, $K_{yy}$, and $K_{zz}$ represent the hydraulic conductivity along the assumed $x$, $y$, and $z$ axes, respectively (m/s); H is the piezometric head (m); and $S_r$ is the volume of water released from storage per unit change in head per unit volume of porous media (m$^{-1}$).

2.4.3. Contaminant Migration Equation

Integrated with Fick's law, the contaminant migration equation is as follows:

$$\begin{aligned}\frac{\partial nC}{\partial t} = &\frac{\partial}{\partial x}\left(nD_{xx}\frac{\partial C}{\partial x} + nD_{xy}\frac{\partial C}{\partial y} + nD_{xz}\frac{\partial C}{\partial z}\right) + \frac{\partial}{\partial y}\left(nD_{yx}\frac{\partial C}{\partial x} + nD_{yy}\frac{\partial C}{\partial y} + nD_{yz}\frac{\partial C}{\partial z}\right)\\ &+ \frac{\partial}{\partial z}\left(nD_{zx}\frac{\partial C}{\partial x} + nD_{zy}\frac{\partial C}{\partial y} + nD_{zz}\frac{\partial C}{\partial z}\right) - \frac{\partial nu_xC}{\partial x} - \frac{\partial nu_yC}{\partial y} - \frac{\partial nu_zC}{\partial z} + I\end{aligned} \tag{3}$$

$D_{xx}$, $D_{yy}$, $D_{xy}$, $D_{yx}$, $D_{xz}$, $D_{yz}$, $D_{zx}$, $D_{zy}$, and $D_{zz}$ are dispersion coefficients in different directions (m$^2$/s); $C$ is the contaminant concentration (mg/m$^3$); $u_x$, $u_y$, and $u_z$ are seepage velocities in the $x$, $y$, and $z$ directions, respectively (m/s); and $n$ is the porosity of the soil.

2.4.4. Simulation Analysis and Verification of Groundwater Flow and Migration

The groundwater flow and Cr(VI) migration model were constructed with Modflow 4.2 software according to above combined conceptualization of the model with groundwater flow and contaminant migration equation [27].

**3. Results**

*3.1. Transport Characteristics of Cr(VI) in Groundwater*

Compared with the breakthrough curve of the conservative tracer Cl$^-$ in Figure 4, the penetration time of Cr(VI) through the groundwater is remarkably longer than that of Cl$^-$. The penetration curve of Cr(VI) (the red line in Figure 4) in the groundwater showed that after Cr(VI) was continuously injected for 20 h, outflowing Cr(VI) ions were detected at the bottom of the soil column. After 48 h of injection, the outflow concentration of the Cr(VI) ions began to gradually increase, and the curve began to warp. After 153 h of injection, the curve became gentle, and the outflow concentration of the Cr(VI) reached 99.5% of the inflow concentration. Hence, the Cr(VI) penetration of the soil column required at least 153 h.

Mathematical analysis for the experiments was performed to model the flow of Cr(VI) through the groundwater matrix to simulate the behavior of Cr(VI) migration in groundwater. When the water level in sampling tubes 1, 2, and 3 remained invariant, the soil column reached saturation, and the water flow in the terminal outlet was stable. At this period, the outflow amount of water in the terminal outlet was about $Q_i$ = 4.5 mL for every $t_i$ = 10 min. The water level in sampling tubes 1 and 3 rose to 565 and 130 mm, respectively. From the center of the soil column, the horizontal distance between sampling tubes 1 and 3 was $\Delta$L = 600 − 200 = 400 mm, and the vertical distance was $\Delta$H = 565 − 130 = 435 mm. The experimental data are listed in Table 1, which shows that the water output in the subsequent experiment stage was kept at 4.5 mL. Thus, this stage can be regarded as a

stable penetration stage, and the hydraulic conductivity ($K_t$) can be calculated using the following formula [28]:

$$K_{t_i} = \frac{10 \cdot Q_i}{S \cdot t_i} \times \frac{l}{\Delta H} \, (mm/\text{min})$$

$$= 0.044 \, mm/\text{min}$$

(4)

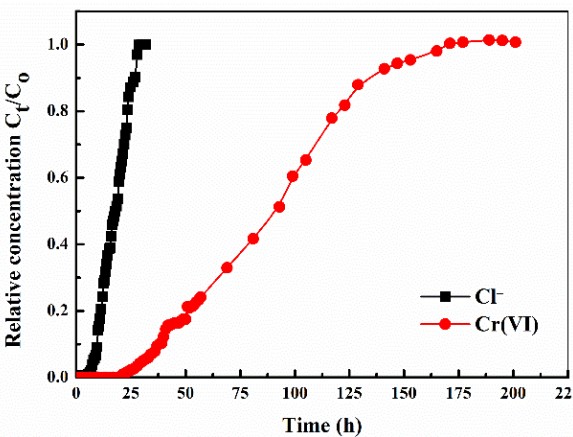

**Figure 4.** Comparison of the breakthrough curves of $Cl^-$ and $Cr(VI)$ in groundwater.

**Table 1.** Soil permeability record.

| ti (min) | Qi (mL) | S (mm) | $V = \frac{10 \cdot Q_i}{S \cdot t_i}$ (mm/min) | Ti (°C) | $K_{t_i}$ (mm/min) |
|---|---|---|---|---|---|
| 0 | 0 | 0 | 0 | 5.3 | 0 |
| 10 | 4.6 | 0.484 | 0.0484 | 5.3 | 0.045 |
| 20 | 4.5 | 0.958 | 0.0474 | 5.4 | 0.044 |
| 30 | 4.5 | 1.432 | 0.0474 | 5.4 | 0.044 |
| 40 | 4.5 | 1.906 | 0.0474 | 5.4 | 0.044 |
| 50 | 4.5 | 2.38 | 0.0474 | 5.5 | 0.044 |
| 60 | 4.5 | 2.854 | 0.0474 | 5.5 | 0.044 |
| 70 | 4.5 | 3.328 | 0.0474 | 5.5 | 0.044 |
| 80 | 4.5 | 3.802 | 0.0474 | 5.5 | 0.044 |
| 90 | 4.5 | 4.276 | 0.0474 | 5.5 | 0.044 |
| 100 | 4.5 | 4.75 | 0.0474 | 5.5 | 0.044 |
| 110 | 4.5 | 5.224 | 0.0474 | 5.5 | 0.044 |
| 120 | 4.5 | 5.698 | 0.0474 | 5.5 | 0.044 |

Given that Cr(VI) was adsorbed by the soil, the adsorption distribution coefficient ($K_d$) and the retardation factor ($R_d$) in a specific soil type should be determined through experimentation. The aforementioned parameters, which were determined through a dynamic adsorption experiment on the soil column, more accurately reflected the migration state than did the balance adsorption experiment. Moreover, the dynamic adsorption experiment requires relatively low costs and time, making it the currently primarily used approach to determining $K_d$ and $R_d$ [29]. According to the adsorption penetration curve of Cr(VI), the $K_d$ and $R_d$ of Cr(VI) were 0.0542 $cm^3$/g and 1.17, respectively.

### 3.2. Model Simulation

3.2.1. Simulation of the Groundwater Flow Field

The groundwater level in the study site is shown in the contour plan of the water line of the groundwater in the evaluated region (Figures 5 and 6). The groundwater flow field (Figure 7) is at 40 to 70 m in the northwest, where the water head is high; in the southeast, the water head is low. The difference between the calculated and measured water heads was verified as less than 3 m. The flow field and flow graph show that the

established groundwater flow field model accurately reflects the relationship among the recharge, flow, and draining of groundwater in the study site. The groundwater accepts recharge primarily from atmospheric precipitation, which is affected by topography. The groundwater basically moves to the Lianshui River in a horizontal direction and to the downstream district along the river and gradually decreases in the vertical direction.

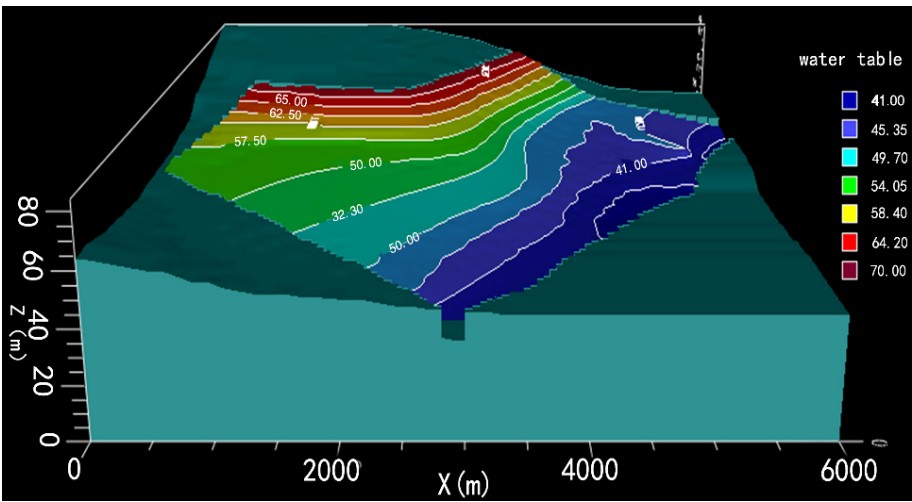

**Figure 5.** Simulation result of the groundwater contour (in meters).

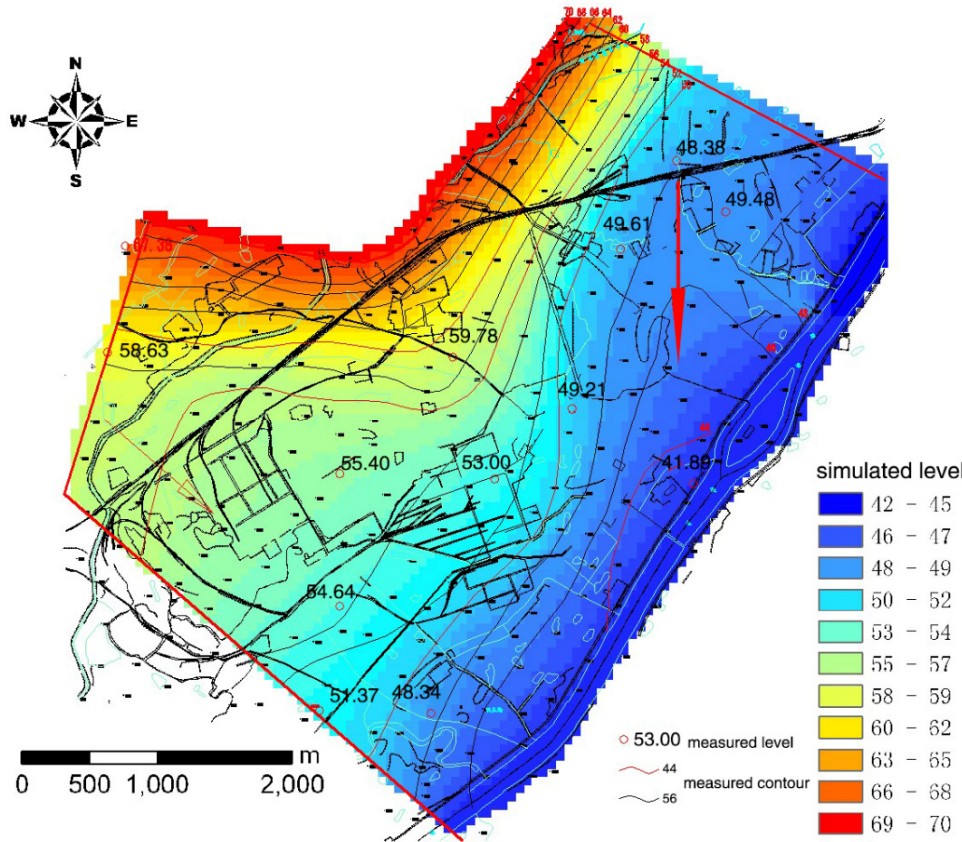

**Figure 6.** Comparison of the simulated groundwater contour with the head surveyed in a real well.

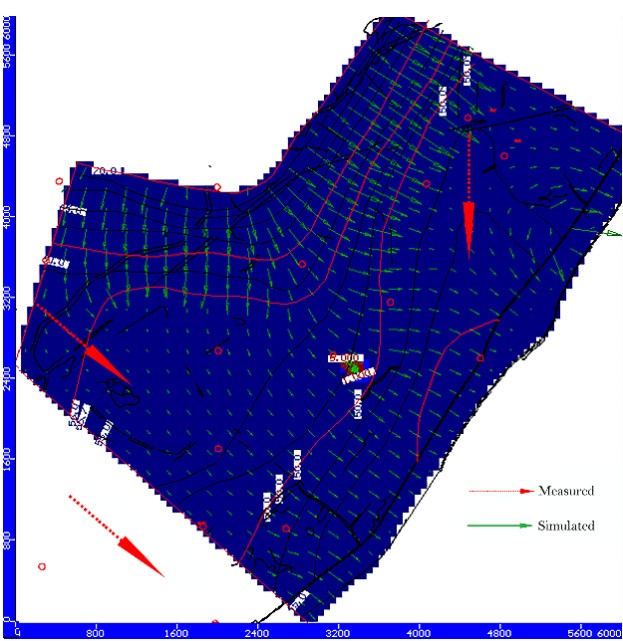

**Figure 7.** Comparison of the simulated flow net with field survey data.

### 3.2.2. Simulation of Cr(VI) Migration

In terms of the basis for determining the flow field, the output of Cr(VI) migration in the groundwater was simulated according to the simulation results for Cr(VI) slag leaching and soil migration, hydrogeological information, and the migration parameters in the experiment. The simulation time was calculated for a certain amount of Cr(VI) slag stockpiling a year after the factory construction; 10 days, 1 year, 2 years, 20 years, and 45 years after the initiation of the simulation were chosen as the simulation output points in the output simulation (Figure 8).

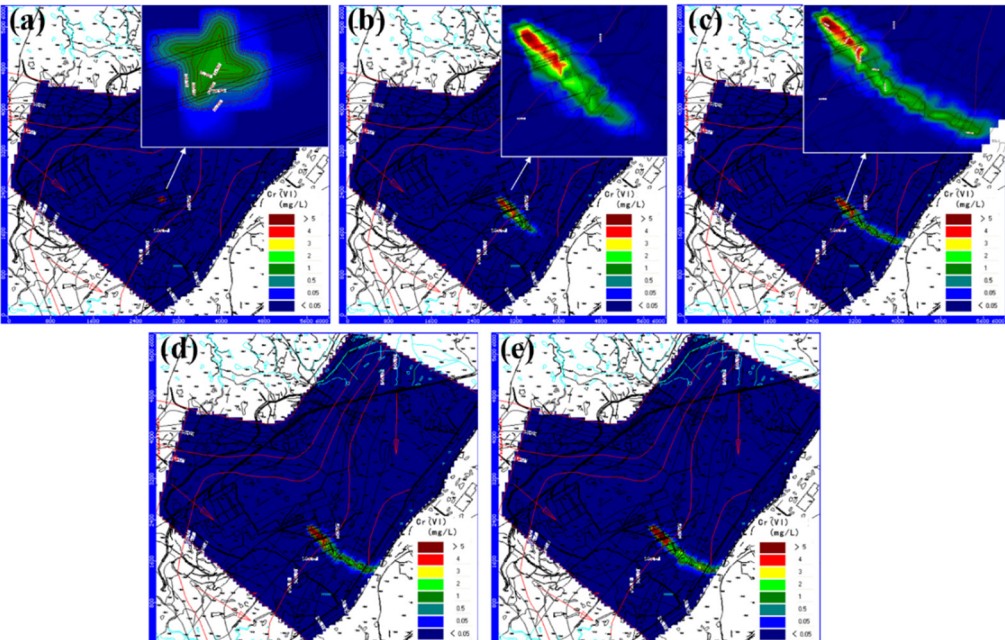

**Figure 8.** Simulation results of Cr(VI) in groundwater after (**a**) 10 days, (**b**) 1 year (1967), (**c**) 2 years (1968), and (**d**) 20 years (1986) since the initiation of the simulation and in (**e**) 2009.

The simulation results showed that the range of Cr(VI) migration in the groundwater gradually increased with time. The pollution halo diffused along the flow direction of the groundwater to the Lianshui River (southeast direction). However, the amount of Cr(VI) that entered the soil and groundwater decreased correspondingly with the increase in Cr(VI) concentrations in Cr leached from the ground slag. Thus, the area of high concentration in the pollution halo had a downsizing trend in the last 25 years of the simulation. During the 45-year period of the simulation, the Cr(VI) pollution halo of the groundwater in the study site continuously spread out to the Lianshui River. Its downstream direction increased the Cr(VI) content of the groundwater in a large area of the study site to a level higher than the standard (0.05 mg/L). Consequently, the water quality in the Lianshui River was considerably affected. Given the presence of wells within the pollution halo, the health of the residents is threatened as well. Notably, the vertical thickness of the aquifuge is small, and Cr(VI) reached the bottom of the aquifer before the simulation was initiated. The vertical difference is imperceptible; thus, the Cr(VI) vertical distribution graph in the groundwater is not presented in this paper.

Sampling analysis for the well water of the residents near this enterprise was performed for the second half of 2009 to verify the reliability of the simulation results. The results were compared with the predicted values for 2009 (Table 2). Table 2 shows that the maximum and minimum differences between the measured and predicted values were 1.158 and 0.001 mg/L, respectively. The maximum relative error (RE) was 31%, the minimum RE was 12%, the average error (ME) was 0.221, and the root mean square error (RMSE) was 0.430. As indicated by these results, the measured and predicted values basically correspond. Therefore, the established mathematical model in this paper is reliable.

**Table 2.** Comparison of simulation and measured values.

| Measured Value of Cr(VI) Concentration (mg/L) | Predicted Value of Cr(VI) Concentration (mg/L) | Relative Error (%) | Measured Value of Cr(VI) Concentration (mg/L) | Predicted Value of Cr(VI) Concentration (mg/L) | Relative Error (%) |
|---|---|---|---|---|---|
| 0.004 | 0.003 | 25 | 4.170 | 3.012 | 28 |
| 0.205 | 0.181 | 12 | 1.774 | 1.225 | 31 |
| 0.032 | 0.026 | 18 | 2.720 | 2.263 | 17 |
| 0.046 | 0.052 | 13 | 0.004 | 0.003 | 25 |
| 0.066 | 0.048 | 27 | 0.004 | 0.003 | 25 |

The predicted values were slightly lower than the measured values because the model simplifies actual situations (Table 2). After treatment, the Cr(VI)-containing wastewater was drained to the Lianshui River from the third branch of the enterprise in Hunan, thereby satisfying the standards. However, the total amount of discharged Cr(VI) in the wastewater still reached 6170 kg each year (1991 statistical data). The results of our sampling and analysis of wastewater from the drainage outlet in 2008 showed that the Cr(VI) contents in all the samples were over 6.49 mg/L, which is 13 times higher than the threshold value (0.5 mg/L) of the discharge standard (GB 8978-1996) for national industrial wastewater. The seasonal lateral recirculation of the Lianshui River supplies the groundwater in urban areas during flooding. Hence, Cr(VI) in the river water may also worsen groundwater pollution. In addition, the breakage and leakage of sewage pipes, as well as Cr(VI) contamination by other manufacturers (e.g., Xiangxiang Tannery), may also discharge Cr(VI) pollutants. These phenomena were not considered in this study. The seasonal changes in the surface water level, the discharge of Cr(VI) pollutants in the surface water, and the possible existence of other pollution sources were also not considered. The exclusion of these factors produced predicted values that were slightly lower than the measured values. However, the findings on the evaluation of the migration route, diffusion range, and pollution degree of pollutants are reliable.

*3.3. Prediction of Different Control Programs for Cr(VI) Contamination of Groundwater*

The migration of Cr(VI) in the groundwater was predicted according to the validation of the reliability of the established model. This model was used to predict the effect of three different control programs for Cr(VI) contamination of groundwater. First, Cr(VI) slag was not processed, and the production was kept in its original status. Second, the slag and over 50% of the Cr(VI) residue was treated in a completely harmless manner. Third, no new slag occurred, and the Cr(VI) slag was treated completely harmlessly. In the three control programs, the migration of Cr(VI) in 2019, 2040, and 2060 was analyzed and predicted (Figure S1).

The results of the prediction for program 1 showed that although all the leached Cr(VI) amounts and concentrations from the old slag decreased, the amount and concentration of leached Cr(VI) still increased because of the continuous merging of new slag. Thus, the pollution halo continuously expanded, and the area of the high-concentration region gradually increased. Therefore, Cr(VI) slag should be treated in a completely harmless manner to cut off the Cr(VI) pollution source of the groundwater.

The prediction for program 2 showed that even in 2060, the range of Cr(VI) pollution halo in the groundwater will not present a downsizing trend. However, the high concentration of Cr(VI) will significantly decrease after five years of the harmless treatment of Cr(VI) slag (2019). After a harmless treatment of 50% of the Cr(VI) slag, the concentration value of Cr(VI) in the groundwater will decrease, and the quality of the groundwater will improve.

The effect of the harmless treatment of Cr(VI) slag can be observed in greater detail in program 3. The area of high Cr(VI) concentration is more obviously reduced, and the highest concentration is not higher than 3 mg/L. By 2060, for most regions before Cr(VI) slag is treated, the highest concentration of Cr(VI) in groundwater will be 10 to 40 times higher than the concentration in the same period after a completely harmless treatment. Therefore, the harmless treatment of Cr(VI) slag significantly improves the groundwater environment of the surrounding regions. Nevertheless, the results of the analysis of program 3 showed that even after 50% of the Cr(VI) slag is treated completely harmlessly, the concentration in a large area with Cr(VI) content still exceeds the groundwater quality standards (GB/T 14848-1993) and the sanitary standard for drinking water (GB5749-2005) of China.

All of these predictions showed that Cr(VI) pollution is a long-term problem. Moreover, the natural repair of groundwater pollution may take a long time. Resolving the pollution problem and restoring the groundwater to its original conditions necessitate substantial effort and investment. However, certain studies have indicated that even with huge investments, such restoration will remain difficult [30]. If no radical measures are taken to control the Cr(VI) pollution caused by the enterprise in this case, the Cr(VI) pollution of the groundwater in Xiangxiang City is expected to worsen for the next several years.

A hypothesis simulation and prediction (program 4) for high concentrations of Cr(VI) pollution in the groundwater were conducted to accurately predict the range of local pollution caused by the factory. The hypothesis is that the pollution concentration of the slag field is 3000 mg/L in the program. The results of the simulation (Figure 9) show that the polluted area considerably expands to the city area in the entire southeast region with the increase in Cr(VI) concentration. The influence of river water causes Cr(VI) to constantly migrate downstream of the river. This migration pollutes the groundwater in most areas of the city, consequently affecting local production and daily life. Accordingly, the source of the pollution problem should be eliminated to improve the quality of the local environment.

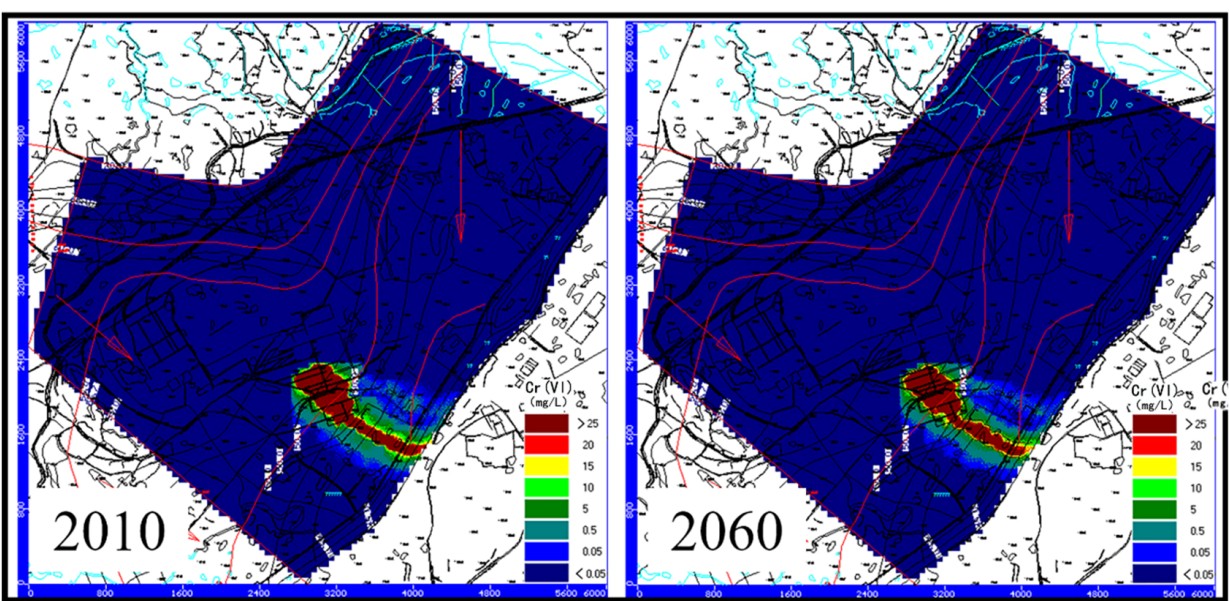

**Figure 9.** Simulation results for Cr(VI) in groundwater with heavy pollution (scenario IV).

## 4. Conclusions

In this paper, the migration mechanism of Cr(VI) leaching from ground slag was analyzed. The Cr(VI) migration process was simulated using an established Cr(VI) migration model. Groundwater pollution hazards were then predicted. Some control programs for Cr(VI) contamination were put forward. These programs have important theoretical and practical significance for the scientific management of Cr(VI) slag and the protection of soil and groundwater environments.

(1) The groundwater level in the study area was 40 to 70 m. The water head was high on the northwest side but low on the southeast side. The difference between the predicted and measured water head values was less than 3 m. In addition, the flow field distribution in the groundwater simulated by the model and the actual situation correspond. The maximum and minimum differences in Cr(VI) between the measured and simulated values were 1.158 and 0.001 mg/L, respectively. The maximum RE was 31%, the minimum RE was 12%, the ME was 0.221, and the RMSE was 0.430. As indicated by the analysis results, the measured and simulated values correspond with each other. Therefore, the established mathematical model in this paper is reliable.

(2) The prediction model shows that the total amount of leached Cr(VI) and the concentration of Cr(VI) slag present a rising trend. The pollution halo continuously expands, and the high-concentration region gradually increases. Therefore, Cr(VI) slag should be treated completely harmlessly to cut off the source of the Cr(VI) pollution.

(3) If the Cr(VI) slag is treated completely harmlessly, the simulation and prediction indicate that the high-concentration area will significantly decrease and that the highest concentration will not be higher than 3 mg/L. By 2060, the highest concentration of Cr(VI) in most of the regional groundwater will be 1/10 and 1/40 of the levels before treatment. Thus, the harmless treatment of Cr(VI) slag considerably improves the quality of groundwater in the surrounding areas.

**Supplementary Materials:** The following supporting information can be downloaded at: https://www.mdpi.com/article/10.3390/pr10112235/s1, Figure S1: Simulation results for Cr(VI) in groundwater under three scenarios: (**a**) untreated slag (scenario I), (**b**) 50% slag treatment (scenario II), and (**c**) 100% slag treatment (scenario III).

**Author Contributions:** Writing—original draft preparation, X.W.; writing—review and editing, T.Y., C.X., and C.L.; Investigation, K.L., R.H., and H.W.; Project administration, Z.W., and Z.Y. All authors have read and agreed to the published version of the manuscript.

**Funding:** This work was supported by the National Natural Science Foundation of China (no. 51204074), Pearl River S&T Nova Program of Guangzhou, China (no. 201710010065), and Science and Technology Innovation Guidance Project of Zhaoqing City (no. 2021040302005).

**Data Availability Statement:** The data presented in this study are available on request from the corresponding author.

**Acknowledgments:** This work acknowledges the Guangdong Provincial Key Laboratory of Environmental Health and Land Resource, Central South University and Zhaoqing University Innovative Research Team of "Water Environment Health Research Team".

**Conflicts of Interest:** The authors declare no conflict of interest.

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
