# Peer review of "Experimental and Modeling Study on Cr(VI) Migration from Slag into Soil and Groundwater"

_processes, doi:10.3390/pr10112235_

Round 1
Reviewer 1 Report
The article titled Experimental and modeling study on Cr(VI) migration from slag into soil and groundwaters was submitted to the journal of Processes. The article described the interesting topic of Cr(Vi) contamination of the environment through slag in the combination with groundwater. The article is interesting and also authors have written a nice introduction into the topic with emphasis their contribution to the topic (well done). The topic itself is appropriate for the selected journal. Due to some minor flaws, the article is suggested for minor revision. The suggestions/comments for authors are listed below.
General remarks:
Please correct all the typos in the text.
Please mark the features on the figures.
Please unify the style of writing, example Cr ore or chromium ore.
Specific remarks:
What is the pH and chemical composition of the groundwater on the given area (please add)? This is important as such, this also contributed to the chemical dynamic/contamination process in the area.
What do you mean by shallow water (give approximation of the depth during the text, not only in the conclusions).
Please define factor S in equation and use the upper or lowercase for the symbols, don´t mix it (text/formula/table).
Figure 6: please mark better, because red marking is not readable.
Reviewer 2 Report
Presnted manuscript deal with the prediction of Cr movement by groundwater in the area of Xiangxiang City, China. The manuscript has a standard structure, is readable and can be interesting for readers. However, several points should be consider before publishing:
- line 35: "According to the National Toxicology Program" of which country?
- line 42: please correct: "studies pay have paid greatly"
- line 57: please correct: "groundwater), , and less"
- section 2.1: references for data and information are missing
- Figure 1: reference and legend is missing
- section 2.2: please specified more in detail sieves
- line 105: please add references "According to international and domestic research..."
- Methodology section: model calibration information is missing; information about used version of model is missing
- Figure 9: it is difficult to read, please make smaller cut out of figures
Reviewer 3 Report
Overall, this paper is well-written, but some minor things need to be addressed before being considered for publication.
1. The figure quality needs to be improved. Legends are not visible
2. What analytical technique was used to determine Cr(VI) speciation? What was the detection limit? The significant figure in Table 2 perhaps needs to be rounded off depending on the detection limit.
